# Generating Summaries with Controllable Readability Levels

**Leonardo F. R. Ribeiro**[†]    **Mohit Bansal**[†,‡]    **Markus Dreyer**[†]

[†]Amazon
[‡]UNC Chapel Hill
{leonribe, mobansal, mddreyer}@amazon.com

## Abstract

Readability refers to how easily a reader can understand a written text. Several factors affect the readability level, such as the complexity of the text, its subject matter, and the reader's background knowledge. Generating summaries based on different readability levels is critical for enabling knowledge consumption by diverse audiences. However, current text generation approaches lack *refined* control, resulting in texts that are not customized to readers' proficiency levels. In this work, we bridge this gap and study techniques to generate summaries at *specified* readability levels. Unlike previous methods that focus on a specific readability level (e.g., lay summarization), we generate summaries with *fine-grained* control over their readability. We develop three text generation techniques for controlling readability: (1) instruction-based readability control, (2) reinforcement learning to minimize the gap between requested and observed readability and (3) a decoding approach that uses lookahead to estimate the readability of upcoming decoding steps. We show that our generation methods significantly improve readability control on news summarization (CNN/DM dataset), as measured by various readability metrics and human judgement, establishing strong baselines for controllable readability in summarization.[1]

## 1 Introduction

Summaries convey salient pieces of information and their understanding depends on the reader's world and domain knowledge. The readability of a text plays a crucial role in how easily it can be understood and consumed for learning and education. Higher readability lowers reading efforts and increases the speed for any reader, and it is particularly beneficial for those who lack high comprehension (DuBay, 2004). On the other hand,

**Document:** Children may be drawn to greasy fries and burgers, but fatty foods can reduce the speed at which their brains work, according to a new study. It found that children who ate a diet higher in saturated fats and cholesterol had slower reaction times and a poorer working memory. Children who ate the fatty diet performed worse when they were given a task-switching game to complete, the researchers said. [...] For the study, scientists at the University of Illinois recruited 150 children aged between seven and 10 and gave them a game which involved learning a pattern between a set of shapes and colours. The game was designed to test cognitive flexibility [...]

**Summarize with readability level 90:** Fatty foods can reduce the speed at which children's brains work. Children who ate a diet high in fat performed worse in a game. This is because it disrupts the way muscles break down sugar in the blood.

**Summarize with readability level 70:** Children who ate fatty and high fat diet had slower reaction times. They performed worse when given a task-switching game to complete. Researchers at Virginia Tech claim that eating fatty food for just five days can change the body's metabolism for the worse.

...

**Summarize with readability level 30:** University of Illinois researchers found children who ate a diet higher in saturated fats and cholesterol had slower reaction times and a poorer working memory. Children were given a game involving learning a pattern between a set of shapes and colours to complete, which tested their cognitive flexibility.

Figure 1: Summaries generated with different readability levels using our lookahead method (Sec. 3.3). We requested summaries with Flesch Reading Ease (FRE) readabilities of 90, 70, 30 (corresponding to the levels of 11-year-old, middle-school and college, respectively). The readability scores of the generated summaries (87.1, 70.3, and 30.9) are close to the requested targets.

lower readability favors specificity, clarity and accuracy (August et al., 2023). Therefore, the readability of a summary is important to ensure that the information is comprehensible to a wider audience, accommodating varying levels of knowledge and understanding (Pitler and Nenkova, 2008).

Significant progress has been made in abstractive summarization using large language models (LLMs) (Raffel et al., 2020; Zhang et al., 2020a; Goyal et al., 2022a). This approach involves making various generation decisions, such as determining which content to paraphrase and how specific a summary should be. The goal is to generate high-quality summaries that are cohesive, readable, and factually consistent. Nevertheless, current methods provide limited mechanisms to specify stylistic preferences such as readability (Goyal et al., 2022b). While readability assessment, which measures the level of difficulty to comprehend a

---

[1]Code/data: https://github.com/amazon-science/controllable-readability-summarization

text, is a well-established field within NLP (Feng et al., 2010; Vajjala, 2022), the control of readability in natural language generation (NLG) tasks, such as summarization, has not been extensively explored and current readability control performance is low (Luo et al., 2022; Pu and Demberg, 2023).

While previous work (Goldsack et al., 2022; Guo et al., 2021; Luo et al., 2022) focused on *binary* readability control, such as expert versus lay summaries, we instead focus on generating summaries of various *fine-grained* reading grade levels (Todirascu et al., 2016; Martinc et al., 2021). Figure 1 presents an example of three summaries with diverse educational levels of readability generated with our lookahead method (Sec. 3.3), given the same document. While the easier-to-understand summary, with the highest Flesch Reading Ease score (Kincaid et al., 1975), uses simpler words, shorter sentences, and less specialized knowledge, the summary with the lowest score requires the reader to understand more complex sentence structures and words (e.g., "saturated", "cholesterol", "cognitive") and contains more specific details (e.g., "University of Illinois researchers"), assuming the readers' familiarity with necessary domain and world knowledge.

We study three methods to control the readability of generated summaries: First, we present an **instruction-prompting** method that prompts the model to output summaries with particular target readability scores or categories, enabling fine-grained control. Next, we develop an approach based on **reinforcement learning (RL)**, using Proximal Policy Optimization (PPO; Schulman et al. (2017)) with a novel Gaussian-based reward that strongly penalizes significant variations in readability, optimizing for summaries with the desired readability. Finally, inspired by Wan et al. (2023), we propose a readability-aware **lookahead decoding** method that selects tokens at each decoding step based on the readability of expected future token generations.

Our contributions in this paper are: (1) We propose three readability-controllable methods for text generation, using instruction-prompting, RL, and lookahead decoding, and (2) show that readability can be explicitly controlled for generating abstractive summaries with finely adjustable levels. Finally, (3) we explore the relation between summary readability and aspects such as specificity, abstractiveness, factuality and informativeness (e.g., more readable summaries tend to have lower specificity and informativeness). Our results show considerable improvements over GPT3.5, a state-of-the-art approach for controlling the readability level (Pu and Demberg, 2023).

## 2 Task Definition: Summaries with Distinct Readability Levels

### 2.1 Task Statement

Readability refers to the ease of understanding text, impacting the reader's comprehension and engagement. We aim to generate summaries with specified levels of readability, given an input document. Let $\boldsymbol{x} = \langle x_1, \ldots, x_n \rangle$ denote the input document represented by the sequence of $n$ tokens, and $\boldsymbol{y} = \langle y_1, \ldots, y_m \rangle$ denote the summary token sequence of length $m$, where $m \ll n$. Let $\hat{r}$ denote the desired summary readability level, which can be represented by a score (Sec. 2.2) or a category name (e.g., "college level", Sec. 3.1). The following formulation represents this task as an auto-regressive problem:

$$p(\boldsymbol{y} \mid \boldsymbol{x}, \hat{r}) = \prod_{i=1}^{m} p(y_i | y_{1:i-1}, \boldsymbol{x}, \hat{r}) \qquad (1)$$

This task presents challenges due to multiple factors: While the method must determine salient information from the input document $\boldsymbol{x}$ and compress it into a concise summary $\boldsymbol{y}$, it must also be able to understand different readability levels and adapt the summarization output to match the target level $\hat{r}$. The approach must strike a balance between providing a succinct, informative summary and ensuring it aligns with the reader's literacy skills and background knowledge (Collins-Thompson, 2014; August et al., 2023).

### 2.2 Readability Metrics

We now discuss metrics to assess text readability. Generally, readability is affected by lexical and syntactic sophistication, sentence structure, discourse cohesion, background knowledge and use of technical terms (McNamara et al., 2010; Crossley et al., 2023). Readability metrics typically focus on syntactic complexity, measuring the presence of qualified syntactic phrases, and vocabulary complexity, aligning words from the text correlated with a particular age-related level (Beers and Nagy, 2009; Crossley et al., 2017). We explore multiple established metrics to control and evaluate

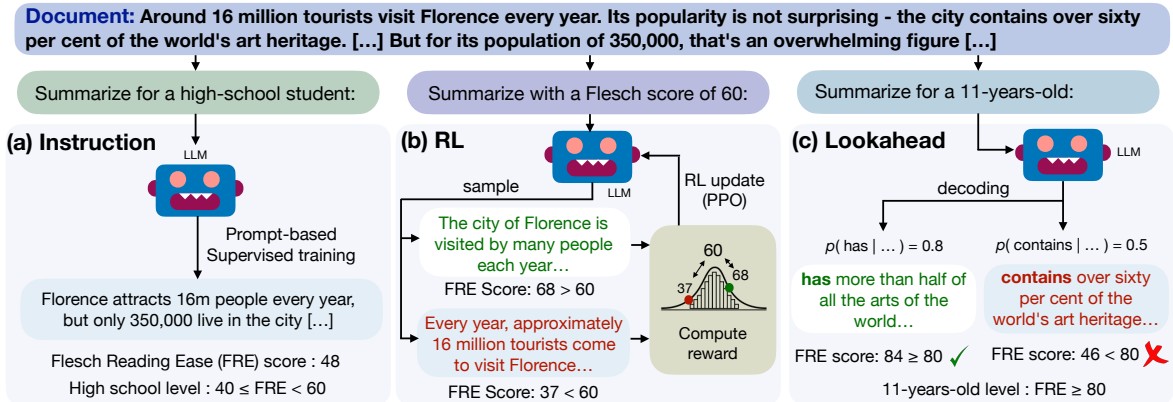

Figure 2: Overview of the proposed methods. (a) illustrates our approach to control the summary readability via fine-grained instructions. (b) shows our RL method where given an input document and the readability level, the policy generates a summary to be scored by our Gaussian-based reward, and (c) shows the our lookahead approach which uses a readability score of a future summary to guide the generation.

the readability of the summaries. Specifically, we employ Flesch Reading Ease (FRE, Kincaid et al., 1975), Gunning fog index (GFI, Gunning, 1952) and Coleman-Liau index (CLI, Coleman and Liau, 1975), among others.[2] Those metrics calculate an approximation of the (US) grade level of education expected to understand a written text.

FRE and GFI metrics are determined by sentence lengths, number of (complex) words, and syllables. Alternatively, unlike syllable-based readability indices, CLI does not require that the character content of words be analyzed, only their length in characters measures readability. Higher FRE scores denote higher readability; higher GFI and CLI scores denote lower readability. The metrics formulas are described in Appendix A. Finally, readability formulas may fail to consider significant factors such as cohesiveness and macro-level organization, which affect the overall readability and understanding of a text (Tanprasert and Kauchak, 2021). To account for this, we measure other aspects such as coherence and informativeness.

## 3 Controllable Methods for Readability

In what follows, we present our three approaches that employ readability metrics (Sec. 2.2) for controllable summary generation, based on fine-grained instructions (Sec. 3.1), reinforcement learning (RL, Sec. 3.2), and lookahead decoding (Sec. 3.3). Figure 2 illustrates the methods.

### 3.1 Instruction-Aligning Readability Methods

Inspired by previous works (He et al., 2022; Zhang and Song, 2022) that explore prompt guidance

[2]Appendix A presents additional readability metrics.

| Readability | Instruction |
|---|---|
| FRE ≥80 | Summarize this for a 11-year-old student: |
| 60≤ FRE <80 | Summarize this for a middle school student: |
| 40≤ FRE <60 | Summarize this for a high school student: |
| FRE <40 | Summarize this for a college student: |

Table 1: Category-based instructions based on readability scores (FRE).

to generate text with desired attributes, we develop instructions that encode the summary readability level. During training, the instructions are prepended to the source documents to create the input for their corresponding summaries. In contrast to recent studies that generate summaries with only two levels (expert and plain language) (Goldsack et al., 2022; Luo et al., 2022), we control the readability using *fine-grained* instructions based on the desired readability, as shown in Figure 2a.

**Category-based Instructions.** Drawing on established guidelines for text complexity levels (Fountas and Pinnell, 1999; DuBay, 2004), we define four instructions based on distinct reading level categories (see Table 1) aligned with particular FRE scores (Vajjala, 2022). For example, we instruct the model to summarize the input document "for a high-school student". We perform instruction-based fine-tuning, selecting the one instruction per training sample that matches the given reading level of the reference summary; that way, the model can learn to associate the observed reference summaries with the categories in the instruction prompts. At inference time, we can select any of the four instructions to request a summary in the style of the specified reading level category. We call this method CATEGORYINSTRUCT.

**Score-based Instructions.** We define a second instruction-based method, which we call SCORE-INSTRUCT. Here, we instruct the model to summarize with a particular score $\hat{r}$, rather than a category. For example, we instruct the model to summarize the input document "with a readability level of 62". In supervised fine-tuning, we use as $\hat{r}$ in the instruction the exact score of each reference summary; this way, the model can learn to associate the observed summaries with the exact reading levels specified in the instructions. At inference time, we can request any score $\hat{r}$. Compared to CATEGORYINSTRUCT, this method adds flexibility by avoiding the hard boundaries presented by readability categories (Table 1); a training sample whose reference summary falls between two reading level categories (e.g., a FRE score of 60 is at the boundary between high-school and middle-school level) does not need to be forced into one or the other category.

### 3.2 Reinforcement Learning for Readability Control

During supervised instruction fine-tuning, using CATEGORYINSTRUCT or SCOREINSTRUCT, we request certain readability levels in the prompts (i.e., readability categories or scores), and the reference summaries act as demonstrations of summaries of that readability level for the model to learn. However, that supervised learning phase does not explicitly check if the model indeed tends to generate summaries in the requested reading levels. It merely uses token-wise gradient updates using teacher forcing based on the reference summaries.

Ouyang et al. (2022) have shown that it can be helpful to supplement such an initial instruction fine-tuning approach with a subsequent reinforcement learning phase, in which the model is further updated in response to sequence-based rewards from a reward model. In contrast to Ouyang et al. (2022), who learned a reward model based on human preferences, we define a reward function based on a readability metric. Intuitively, we want to reward the model maximally whenever it generates a summary that has the requested readability level $\hat{r}$ and decrease the reward steeply for generated summaries whose readability deviates from $\hat{r}$.

**Reward.** We design a reward $R(\tilde{r}_{\boldsymbol{y}}, \hat{r})$ that assigns a maximum reward of 1.0 if the observed readability $\tilde{r}_{\boldsymbol{y}}$ of a generated summary $\boldsymbol{y}$ is equal to the desired readability $\hat{r}$ and decreases exponentially as $\tilde{r}_{\boldsymbol{y}}$ deviates from $\hat{r}$. As illustrated in Fig-

ure 2b, we formulate it as a normalized Gaussian centered at $\hat{r}$:

$$f = \mathcal{N}(\hat{r}, \sigma^2) \qquad (2)$$
$$R(\tilde{r}_{\boldsymbol{y}}, \hat{r}) = f(\tilde{r}_{\boldsymbol{y}})/f(\hat{r}), \qquad (3)$$

The use of a Gaussian function ensures that the reward decreases in a nonlinear fashion: Small readability deviations from the requested readability $\hat{r}$ result in small reward reductions; larger readability deviations result in disproportionally larger reward reductions. This is analogous to the nonlinear penalties for deviations from a Gaussian prior in L2 weight regularization.

**PPO.** We apply RL using the described reward, aiming for the model to learn improved readability control. At the same time, we wish to preserve the existing high salience and coherence achieved by the summarization models tuned with supervised fine-tuning. To accomplish this, we initialize the RL policy with SCOREINSTRUCT that has been trained on supervised data. We then employ the popular policy gradient method PPO (Schulman et al., 2017), which has been successfully applied to text generation problems (Liu et al., 2022a) and has been shown to be sample efficient and stable (Wu et al., 2021). To ensure stability, PPO employs a clipping mechanism during policy updates, preventing drastic changes and avoiding the problem of diverging policy updates. We call the resulting PPO-tuned model SCOREINSTRUCT+RL.

### 3.3 Lookahead Readability Decoding

As a direct consequence of RL training, the model explicitly learns to generate tokens with the goal of maximizing the readability reward. However, RL requires a initialization with a high-quality supervised model, which might not always be available. Consequently, we explore an decoding approach to dynamically adapt the readability level during *inference*, as shown in Figure 2c.

Previous work (Lu et al., 2022; Wan et al., 2023) develop lookahead approaches for controlling generation through signals such as logic-based lexical constraints or faithfulness. We extend this concept to enhance *readability control* in abstractive summarization at inference time. We develop a lookahead strategy that directs the model generation process, increasing the likelihood that the chosen tokens will lead to a path with the intended level of readability in the search space. Formally,

each summary token $y_i$ is selected by:

$$g(y_i) = \log p(y_i \mid y_{1:i-1}, \boldsymbol{x}, \hat{r})$$
$$+ w \cdot \max_{L_y} h(y_{1:i-1+n}, \hat{r}) \qquad (4)$$

where $h(\cdot)$ is a function that assigns a readability score to the summary and $w$ controls the weight of the readability in the generation process. $L_y$ is a set of possible future summaries that start with the tokens $y_{1:i-1}$ and contains $n$ additional continuation tokens likely to be encountered in the future. Our readability evaluation function $h$ is defined as:

$$h(\boldsymbol{y}, \hat{r}) = 1 - |\tilde{r}_{\boldsymbol{y}} - \hat{r}| \qquad (5)$$

where $\tilde{r}_{\boldsymbol{y}}$ is the observed readability of the (possibly incomplete) summary $\boldsymbol{y}$. $h(\cdot)$ can be defined with additional metrics which can be combined to desired scores (see Sec. 5.4). Finally, note that this method is computational costly since it needs to generate future summaries per generation step.

## 4 Experimental Setup

We evaluate on CNN/DM (Hermann et al., 2015), which contains news articles and their bullet-point highlights considered as summaries. In line with previous work (Guo et al., 2023) where summaries have similar or easier readability level than the input document, we assess the subset of the CNN/DM test set with documents with a FRE score below 50 (high school and college student levels).

The readability of the summaries are computed using FRE metric and used during training for constructing the instructions for CATEGORYINSTRUCT and SCOREINSTRUCT, and as the desired readability $\hat{r}$ in SCOREINSTRUCT+RL and SCOREINSTRUCT+LA. For CATEGORYINSTRUCT, we map the FRE score to the instruction for each reference summary as shown in Table 1. In SCOREINSTRUCT+RL, we randomly sample FRE scores which are used in the instruction to generate the summary and as $\hat{r}$ in the reward. We define $\sigma$ by drawing inspiration from the readability levels (see Table 1), with each level covering a range of 20 FRE points. We set the standard deviation $\sigma$ to half of that value (i.e., 10), so that more than two thirds of the reward values lie within the requested reading level centered at $\hat{r}$. In SCOREINSTRUCT+LA, the number of tokens considered in future $n$ is set to 20.

Our models are initialized with Flan-T5-Large (Chung et al., 2022) and fine-tuned using a context size of 1024. We include as baselines GPT3.5 (TEXT-DAVINCI-003) and a Flan-T5

|  | FRE$\Delta$ $\downarrow$ | FRE$\rho$ $\uparrow$ | GFI$\rho$ $\downarrow$ | CLI$\rho$ $\downarrow$ |
|---|---|---|---|---|
| GPT3.5 (TEXT-DAVINCI-003) | 24.2 | 0.24 | -0.28 | -0.16 |
| Best-GPT3.5 (TEXT-DAVINCI-003) | 18.4 | 0.59 | -0.50 | -0.45 |
| CATEGORYINSTRUCT | 15.8 | 0.59 | -0.37 | -0.55 |
| SCOREINSTRUCT | 15.1 | 0.62 | -0.40 | -0.58 |
| SCOREINSTRUCT+RL | 11.8 | 0.74 | -0.42 | -0.72 |
| SCOREINSTRUCT+LA | **4.5** | **0.95** | **-0.67** | **-0.88** |

Table 2: Readability results for the proposed methods using different automatic metrics. $\Delta$ is the absolute difference between the desired and generated FRE readability. $\rho$ is the correlation with the target metric. All correlations are statistically significant (p<0.05).

model without readability control. As an additional baseline, which we call Best-GPT3.5, we sample $k$ summaries from GPT3.5 for each instruction and select the summary whose readability level is closest to the requested level.[3]

We evaluate the readability with the FRE, GFI and CLI metrics (Sec. 2.2). We use BertScore (BS, Zhang et al., 2020b) and the F1 measure of ROUGE-L (RG-L, Lin, 2004) for evaluating the summary quality; and FactCC (Kryscinski et al., 2020) and UniEval (Zhong et al., 2022) for faithfulness assessment.

## 5 Results and Analysis

### 5.1 Main Readability Results

Table 2 shows the results on CNN/DM summaries, where we compute the absolute difference between the desired and generated FRE readability ($\Delta$) and Pearson correlations with readability metrics. In comparison to our methods, GPT3.5 summaries are considerably distant from the intended reading levels, while Best-GPT3.5 greatly improves over GPT3.5. SCOREINSTRUCT has better performance than CATEGORYINSTRUCT, suggesting that fine-grained readability signals are beneficial during training. SCOREINSTRUCT+RL enhances readability control significantly, while SCOREINSTRUCT+LA further improves all readability metrics.

Table 3 presents the detailed results by reading level.[4] To compare SCOREINSTRUCT's methods with GPT3.5 and CATEGORYINSTRUCT, we used prompts with the specific values for each readability level (30, 50, 70 and 90). Both instruction methods generates summaries with granular readability measured by the 3 readability metrics, while maintaining the summary quality close to the baseline.

For SCOREINSTRUCT+RL, the largest gains are for the *11-year-old* level, improving FRE from 67.6 to

---

[3]Hyper-parameters and prompts are found in Appendix B.
[4]Other readability metrics are presented in Appendix D.

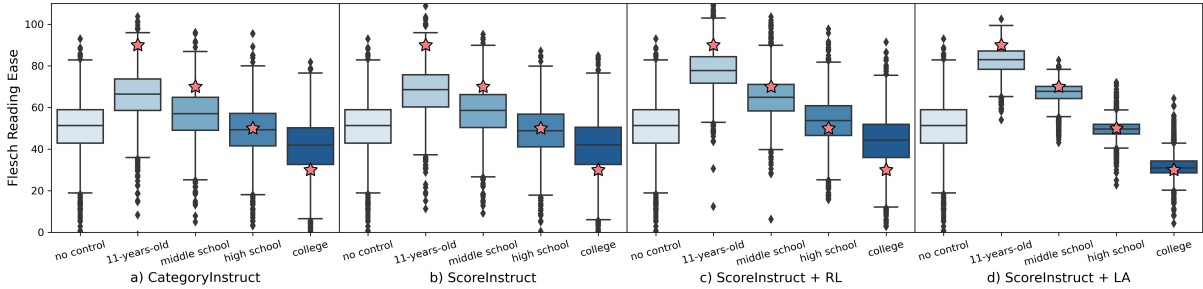

Figure 3: Readability scores for summaries generated from CNN/DM test set. The stars indicate the requested readability level.

| Readability ($\hat{r}$) | Readability ($\tilde{r}_y$) | | | | Quality | |
|---|---|---|---|---|---|---|
| | FRE↑ | FRE∆↓ | GFI↓ | CLI↓ | BS↑ | RG-L↑ |
| No control | 50.1 | - | 10.9 | 11.6 | 0.881 | 38.73 |
| **GPT3.5** (TEXT-DAVINCI-003) | | | | | | |
| 11-year-old | 44.2 | 45.7 | 13.3 | 12.3 | 0.872 | 31.07 |
| Middle school | 41.5 | 28.4 | 14.0 | 12.6 | 0.872 | 31.38 |
| High school | 39.1 | 12.3 | 14.7 | 12.9 | 0.871 | 31.18 |
| College student | 37.0 | 10.3 | 15.3 | 13.2 | 0.870 | 31.19 |
| **Best-GPT3.5** (TEXT-DAVINCI-003) | | | | | | |
| 11-year-old | 50.9 | 39.0 | 12.3 | 11.2 | 0.871 | 31.41 |
| Middle school | 47.8 | 22.2 | 13.1 | 11.7 | 0.871 | 31.74 |
| High school | 43.9 | 7.0 | 14.0 | 12.2 | 0.870 | 31.57 |
| College student | 33.4 | 5.4 | 15.9 | 13.7 | 0.868 | 30.69 |
| **CATEGORYINSTRUCT** | | | | | | |
| 11-year-old | 65.7 | 24.3 | 8.5 | 9.1 | 0.879 | 36.87 |
| Middle school | 56.4 | 14.6 | 9.8 | 10.6 | 0.882 | 38.73 |
| High school | 48.7 | 9.6 | 10.9 | 11.9 | 0.882 | 38.67 |
| College | 40.9 | 14.6 | 11.7 | 13.3 | 0.881 | 37.72 |
| **SCOREINSTRUCT** | | | | | | |
| 90 (11-year-old) | 67.6 | 22.5 | 8.1 | 8.7 | 0.878 | 35.38 |
| 70 (Middle school) | 57.7 | 13.6 | 9.6 | 10.4 | 0.882 | 38.34 |
| 50 (High school) | 48.4 | 9.5 | 10.8 | 12.0 | 0.882 | 38.64 |
| 30 (College) | 41.1 | 14.7 | 11.5 | 13.3 | 0.881 | 37.09 |
| **SCOREINSTRUCT+RL** | | | | | | |
| 90 (11-year-old) | 77.8 | 13.3 | 6.7 | 6.9 | 0.856 | 28.60 |
| 70 (Middle school) | 64.7 | 9.0 | 8.5 | 9.3 | 0.868 | 33.21 |
| 50 (High school) | 53.5 | 9.2 | 9.7 | 11.3 | 0.873 | 34.97 |
| 30 (College) | 43.8 | 15.6 | 10.6 | 13.0 | 0.873 | 34.76 |
| **SCOREINSTRUCT+LA** | | | | | | |
| 90 (11-year-old) | 83.2 | **7.1** | 6.6 | 6.3 | 0.868 | 30.75 |
| 70 (Middle school) | 67.1 | **3.8** | 8.5 | 8.9 | 0.876 | 35.05 |
| 50 (High school) | 49.2 | **2.9** | 10.7 | 11.9 | 0.878 | 36.01 |
| 30 (College) | 31.9 | **4.0** | 12.5 | 14.8 | 0.874 | 32.66 |

Table 3: Detailed results on CNN/DM using Instruction-prompting, RL and Lookahead (LA) methods. Higher FRE (↑) denotes higher readability, while lower GFI and CLI (↓) denote higher readability.

77.8. However, the summary quality is affected with decrease on RG-L. SCOREINSTRUCT+LA is very effective further decreasing ∆ over all reading levels. On the other hand, the high computational expense decoding (generating future summaries) is a disadvantage of this method.[5] While the readability of GPT3.5-generated expert-style summaries is lower than easier-to-understand summaries, the readability variation between the different sum-

mary types is significantly less than our methods. Recently, Pu and Demberg (2023) found that the proficiency of GPT3.5 in adapting the readability of summaries is behind human-written texts. Conversely, while selecting a summary with the closest readability level from the sampled GPT3.5 summaries improves the performance, it is time-consuming and costly.

Figure 3 visualizes the variance of our approaches for each readability level, according to FRE, while Figure 4a gives an overview of the relation between observed and requested readability for SCOREINSTRUCT's methods. Generally, while the instruction models can distinguish summaries at different readability degrees, SCOREINSTRUCT+RL is more effective in telling apart the defined levels, while SCOREINSTRUCT+LA is the approach with less variance and stronger fine-grained control.

## 5.2 Impact of Other Summary Dimensions

**Specificity.** Summaries can differ in the level of details they convey, which can range from being highly specific to more general. While easier sentences are less detailed and use simple vocabulary (Laban et al., 2021), casual language (Pitcher et al., 2022), and concise sentences (Scarton et al., 2018), specific sentences are more likely to contain specific words and details. Following Goyal et al. (2022b), we measure the degree of specificity of the generated summaries using Speciteller tool (Li and Nenkova, 2015).[6] Interestingly, we observe in Figure 4b that easy-to-read texts are less specific, while summaries with higher readability are more specific, demonstrating that our methods enable summaries with different specificity degrees.

**Abstractiveness.** Abstractiveness quantifies the extent of rephrasing in generated text, indicating

---

[5]Future work will consider to distillate into a model desired readability levels generated via lookahead (Wan et al., 2023).

[6]The summaries are segmented into sentences, and the macro-average of sentence-level specificity is calculated.

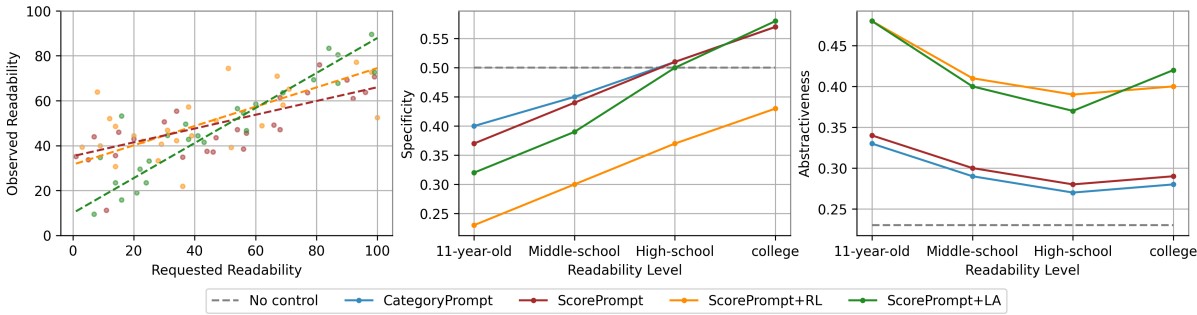

Figure 4: (a) Relation between observed $\tilde{r}_y$ versus requested $\hat{r}$. (b) Specificity and (c) Abstractiveness of the CNN/DM summaries based on four readability levels.

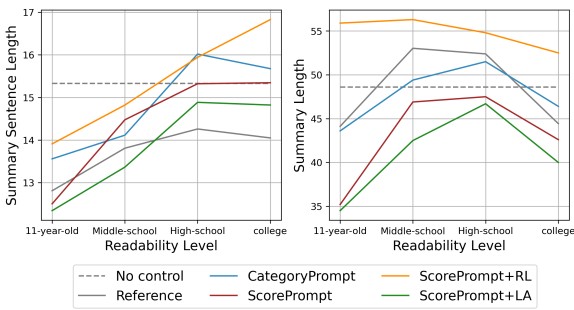

Figure 5: Summary sentence lengths (left) and summary lengths (right) generated by the proposed approaches for different readability levels, compared to reference summaries of the corresponding levels.

the level to which the words are not directly taken from the input. We employ MINT (Dreyer et al., 2023) to measure the abstractiveness based on overlaps between the input document and the summary. As shown in Figure 4c, easier summaries are more abstractive than more complex summaries, and college-level summaries are slightly more abstractive. We hypothesize that this is because most of CNN/DM documents have readability levels in the range of high school, making more likely that the model copy parts from the source with similar level, producing more extractive summaries.

**Summary Lengths.** FRE is sensitive to text lengths (Tanprasert and Kauchak, 2021) and hence we check whether simpler summaries are just shorter while more difficult summaries are longer. Optimizing toward higher FRE scores leads to summaries that contain words that are shorter and easier to read and sentences that are shorter.[7] As shown in Figure 5, we do not observe very short or long summaries overall. Finally, most methods generate summaries that are similar in length to the (training set) reference summaries, while the RL method detaches the model from such length constraints.

---

[7]Appendix F shows summaries with distinct levels.

| Readability ($\hat{r}$) | Readability | | Informativeness | |
|---|---|---|---|---|
| | $\mu_r \uparrow$ | $\sigma_r$ | $\mu_i \uparrow$ | $\sigma_i$ |
| Reference (No control) | 16.10 | 2.40 | 28.02 | 2.11 |
| Baseline (No control) | 18.45 | 2.27 | 30.20 | 2.11 |
| **CATEGORYINSTRUCT** | | | | |
| 11-year-old | 26.97 | 2.16 | 25.55 | 2.16 |
| Middle school | 23.81 | 2.32 | 28.60 | 2.20 |
| High school | 17.94 | 2.33 | 29.14 | 2.10 |
| College student | 17.37 | 2.28 | 30.14 | 2.09 |
| **SCOREINSTRUCT** | | | | |
| 90 (11-year-old) | 33.39 | 2.23 | 21.19 | 2.20 |
| 70 (Middle school) | 23.30 | 2.27 | 29.00 | 2.29 |
| 50 (High school) | 22.83 | 2.45 | 31.88 | 2.22 |
| 30 (College student) | 25.43 | 2.31 | 24.59 | 2.21 |
| **SCOREINSTRUCT+RL** | | | | |
| 90 (11-year-old) | 34.71 | 2.39 | 07.93 | 3.47 |
| 70 (Middle school) | 31.13 | 2.21 | 21.75 | 2.22 |
| 50 (High school) | 29.46 | 2.26 | 23.69 | 2.18 |
| 30 (College student) | 29.99 | 2.20 | 26.14 | 2.28 |
| **SCOREINSTRUCT+LA** | | | | |
| 90 (11-year-old) | 30.41 | 2.25 | 16.28 | 2.38 |
| 70 (Middle school) | 27.91 | 2.29 | 25.03 | 2.24 |
| 50 (High school) | 21.30 | 2.27 | 27.46 | 2.11 |
| 30 (College student) | 22.10 | 2.34 | 25.80 | 2.23 |

Table 4: Human evaluation of readability ($\mu_r$) and informativeness ($\mu_i$) of CNN/DM summaries. $\sigma_r$ and $\sigma_i$ are the corresponding standard deviations.

**Case Study.** Table 5 shows examples of summaries with different readability levels generated by SCOREINSTRUCT+LA.[8] Note that the observed FRE score decreases as the target FRE decreases. Summaries with lowest readability have more specific words (e.g., "defeated", "contest") and provide more details (e.g., "El Clasico", "Brazilian"). Table 11 in the Appendix presents examples generated by the other methods.

### 5.3 Human Evaluation

We conduct human evaluations to determine how readable and informative the CNN/DM generated summaries are, using Amazon Mechanical Turk. Details on setup, annotation instructions and fair compensation are described in Appendix E.

We ask to select the most readable and the least readable summaries among three displayed sum-

---

[8]Readability scores generated using https://github.com/cdimascio/py-readability-metrics.

| | FRE↑ | GFI↓ | CLI↓ |
|---|---|---|---|
| **Document:** Team-mates Neymar and Dani Alves proved their dedication to Barcelona by supporting the club's basketball side. Neymar and Alves headed to watch El Clasico on Thursday night alongside the Brazilian's sister Rafaella. Barca prevailed with a narrow 85-80 victory in the Euro League contest. Brazil star Neymar takes a selfie with friends and Barcelona team-mate Dani Alves However Real Madrid remain top of their Euro League division over their bitter rivals, just by points difference ... Neymar's sister Rafaella headed to watch El Clasico of basketball with the Barcelona forward ... | 45.7 | 11.1 | 10.5 |
| **Requested** FRE 90: Real Madrid and Barcelona played basketball on Thursday night. Barca won the game 85-80, but Real are top of the Euro League by points. Neymar and his sister Rafaella went to watch the game with friends. | 77.1 | 8.1 | 5.9 |
| **Requested** FRE 70: Barcelona beat Real Madrid 85-80 in their Euro League basketball clash. Neymar's sister Rafaella joined him and friends at the game on Thursday. Real Madrid are top of their division, just by points difference. | 65.9 | 9.3 | 8.0 |
| **Requested** FRE 50: Barcelona defeated Real Madrid 85-80 in El Clasico on Thursday night. Neymar and his Barcelona team-mates went to watch basketball with his sister Rafaella. Real remain top of the Euro League table over Barcelona by just points. | 50.2 | 9.8 | 7.1 |
| **Requested** FRE 30: Neymar and his Barcelona team-mates attended an El Clasico basketball game. Barcelona defeated Real Madrid 85-80 in the Euro League contest. The Brazilian forward's sister Rafaella also attended the game. | 33.1 | 12.0 | 9.3 |

Table 5: Examples of summaries of different readability levels generated using SCOREINSTRUCT+LA.

| Readability ($\hat{r}$) | FactCC↑ | UniEval | |
|---|---|---|---|
| | | Consistency↑ | Coherence↑ |
| **SCOREINSTRUCT** | | | |
| 11-year-old | 0.63 | 0.918 | 0.919 |
| Middle school | 0.65 | 0.923 | 0.924 |
| High school | 0.66 | 0.929 | 0.931 |
| College student | 0.65 | 0.933 | 0.931 |
| **SCOREINSTRUCT+RL** | | | |
| 11-year-old | 0.58 | 0.913 | 0.919 |
| Middle school | 0.62 | 0.925 | 0.936 |
| High school | 0.64 | 0.936 | 0.941 |
| College student | 0.62 | 0.939 | 0.940 |
| **SCOREINSTRUCT+LA** | | | |
| 11-year-old | 0.54 | 0.851 | 0.801 |
| Middle school | 0.59 | 0.895 | 0.870 |
| High school | 0.59 | 0.911 | 0.896 |
| College student | 0.54 | 0.901 | 0.864 |

Table 6: Factuality, consistency and coherence results.

maries. Such a relative selection is easy to do, while determining an absolute ordinal readability level per summary would require expert training (Kiritchenko and Mohammad, 2017). We adopt a rating estimation in which game players' skills are estimated based on a series of multi-player games (Weng and Lin, 2011).[9] This is similar to the ELO score (Elo, 1978) used in chess, but it computes rating mean and standard deviation. Ratings start at 25.0 and are adapted based on wins and losses. We draw 1,000 three-summary sets, where the summaries in each set are generated from the same input article and are randomly drawn from the pool of 18 summaries (4 methods x 4 target readability levels, plus baseline and reference) per input article. Therefore, estimated ratings can be compared across different settings. For informativeness, we instruct annotators to mark the *least informative* and the *most informative* summary. Table 4 shows the human evaluation results. For all four methods, the readability tends to increase as higher target readability is requested, confirming

the effectiveness of our methods, while informativeness is negatively correlated with readability.

## 5.4 Factual Consistency and Coherence

Factual consistency and coherence are crucial aspects in abstractive summarization. However, ensuring the factuality and coherence while controlling for readability is challenging. Table 6 shows that in most cases easy-to-read summaries, which are more abstractive (Figure 4c), are less factual and coherent. Previous work (Ladhak et al., 2022; Dreyer et al., 2023) show that factuality decays with increasing abstractiveness. Note that we explicitly used readability signals to control the models towards generating summaries with different readability degrees. However, a high-quality summary should also be faithful and contain relevant information, traits that may not be captured by readability metrics alone. Additionally, a signal based only on readability raises the risk of degenerate solutions, which might result on non-coherent summaries. We are interested in exploring whether factuality metrics in combination with readability affects this dynamics. As shown in Table 7, we adapt the RL and LA methods with a linear combination of BS-Fact (BERTScore precision of a summary with respect to the source document) and FRE readability scores. Note that when optimizing only for readability, FRE $\Delta$ is lower but factual consistency and RG-L are the worst, and using BS-Fact tends to improve the metrics while reducing the readability control. Finally, Table 8 presents the impact of considered future tokens in LA, showing that looking ahead for more tokens in fact improves factuality while decreasing FRE $\Delta$.

## 6 Related Work

**Readability Control in NLG.** Early efforts for readability control in NLG include microplanning

---

[9] We use the implementation in https://openskill.me.

| | FRE $\Delta\downarrow$ | BertScore$\uparrow$ | RG-L$\uparrow$ | FactCC$\uparrow$ |
|---|---|---|---|---|
| **SCOREINSTRUCT+RL** | | | | |
| Only FRE | 11.9 | 0.869 | 33.76 | 0.64 |
| 0.65 FRE + 0.35 BS-Fact | 13.3 | 0.877 | 33.30 | 0.65 |
| 0.35 FRE + 0.65 BS-Fact | 15.2 | 0.879 | 35.42 | 0.64 |
| Only BS-Fact | 15.2 | 0.879 | 35.43 | 0.64 |
| **SCOREINSTRUCT+LA** | | | | |
| Only FRE | 4.89 | 0.874 | 33.81 | 0.56 |
| 0.65 FRE + 0.35 BS-Fact | 6.72 | 0.877 | 36.02 | 0.59 |
| 0.35 FRE + 0.65 BS-Fact | 9.33 | 0.879 | 37.32 | 0.63 |
| Only BS-Fact | 16.23 | 0.882 | 38.72 | 0.68 |

Table 7: Ablation using Readability (FRE) and BS-Fact score combinations in the CNN/DM validation set.

steps to tailor the generated text to match different target reading levels (Moraes et al., 2016) and adapting the text complexity in machine translation outputs (Agrawal and Carpuat, 2019; Marchisio et al., 2019). Recently, Luo et al. (2022) investigate controllable abstractive and extractive approaches for generating layman and expert summaries from biomedical documents. Concurrent with our work, Pu and Demberg (2023) conduct a inspection of the ability of a GPT3.5 model to adapt its output to different target audiences and writing styles (formal vs. informal), while Imperial (2022) found that that GPT2 models struggle in preserving the linguistic complexity of the input prompts. Importantly, there has been significant development of models for Plain Language Summarization (PLS) from scientific papers (Devaraj et al., 2021; August et al., 2022; Goldsack et al., 2023; Guo et al., 2023). However, different from our proposed methods, such works do not consider fine-grained readability degrees. Concurrent to our work, Chi et al. (2023) employ weakly supervision and prompt methods to control readability complexity level of sentence paraphrasing.

**Controllable Text Generation.** Previous work explore different alternatives to tailor text for diverse target users (Cao et al., 2020; Kumar et al., 2022; Pu and Demberg, 2023), while research has also been conducted on style control in various generation tasks, including paraphrasing and story generation (Wang et al., 2017; Shen et al., 2017; Huang et al., 2019). In particular, for summarization, style control emphasizes factors such as abstractiveness (Goyal et al., 2022b; Dreyer et al., 2023) length, or content (Fan et al., 2018; He et al., 2022; Shen et al., 2022; Liu et al., 2022b). Contrary to these, Böhm et al. (2019) employ RL with rewards from human preferences for generating different summaries, while our reward function is based on readability signals. Goyal et al. (2022b)

| Future tokens | FRE $\Delta\downarrow$ | BertScore$\uparrow$ | RG-L$\uparrow$ | FactCC$\uparrow$ |
|---|---|---|---|---|
| 3 | 6.1 | 0.872 | 33.10 | 0.51 |
| 5 | 5.5 | 0.873 | 33.19 | 0.51 |
| 10 | 4.8 | 0.873 | 33.53 | 0.52 |
| 20 | 4.5 | 0.874 | 33.69 | 0.54 |

Table 8: Impact of the number of future tokens ($n$) in SCOREINSTRUCT+LA in the CNN/DM validation set.

extend a single decoder framework to a mixture-of-experts architecture with multiple decoders, allowing the model to learn different summary styles. Zhang et al. (2022) produce summaries using designed attributes such as length and extractiveness. In contrast, in this work, we focus on readability levels in summary generation and propose training and decoding control strategies for their control.

**Text Simplification.** Text Simplification aims to improve the readability of sentences through reducing linguistic complexity while maintaining the original meaning (Alva-Manchego et al., 2021). Different aspects of the simplified output have been controlled, such as adapting to a specific level (Scarton and Specia, 2018; Nishihara et al., 2019) or incorporating edit operations (Alva-Manchego et al., 2017; Kumar et al., 2020; Mallinson et al., 2020) or lexical and syntactic constraints (Martin et al., 2020) into text simplifications. Maddela et al. (2021) implement linguistically motivated syntactic rules with data-driven neural models to enhance the diversity and controllability of the simplifications. In contrast to text simplification, which aims to control the degree to which a sentence is paraphrased, our approaches must provide succinct and informative summaries while maintaining different fine-grained levels of desired readability.

## 7 Conclusion

In this work, we propose three methods for fine-grained control of the readability level of summaries. We showed that instruction-based methods can be used to guide LLMs to generate summaries with fine-grained readability degrees. We thus presented a RL approach that uses a Gaussian-based reward and a new decoding method that allows to control the readability during inference. We provided an extensive evaluation of our approaches and showed that they significantly improves the control of the summaries' readability. Future work includes adapt the methods to different NLG tasks and combine different metrics in order to capture distinct summary aspects that impact readability.

## Limitations

In this paper, we propose different methods to current summarization approaches to enhance the controllability of readability levels. While this adjustment is not specific to any particular language, we conducted all of our experiments and analysis exclusively on English-language summarization datasets. Additionally, we focused solely on studying newswire summaries, given their widespread use in summarization research. Hence, this paper does not offer insights into the range of style variations found in non-English and non-newswire datasets, nor does it ascertain the generalizability of our findings to other datasets and domains. Second, although the Lookahead method demonstrates enhanced readability control, it requires a heavy computational overhead, especially when it is used with larger beams. To alleviate these expenses, one possible solution is to employ distillation to enhance decoding speed (Wan et al., 2023). Finally, the experimental cost of requesting API responses from OpenAI to assess ChatGPT's text generation abilities imposes significant constraints on the dataset selection. In this way, we limit our experimentation to only one summarization dataset.

## Ethics Statement

While some of the investigated systems have demonstrated a high level of controllability on the CNNDM dataset, this does not imply their use as general controllable summarization models. To ensure reliability, these models should be thoroughly evaluated before being used in different settings.

In the conducted human experiment, for informativeness, workers took a median time of 79 seconds per HIT, and we paid $0.40 plus a bonus of $0.10, which amounts to $22.80 per hour. For readability, workers took a median time of 58 seconds per HIT, and we paid $0.20 plus a bonus of $0.05, amounting to a pay of $15.50 per hour.

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

# Appendix

## A  Readability Metrics

Flesch reading ease (FRE) (Kincaid et al., 1975) metric assigns higher scores to texts that are easier to read. It is calculated as follows:

$$\text{FRE} = 206.835 - 1.015(\frac{\text{totalWords}}{\text{totalSentences}}) - 84.6(\frac{\text{totalSyllables}}{\text{totalWords}}).$$

The Gunning fog index (GFI) (Gunning, 1952) quantifies the level of formal education required for a person to comprehend a given text upon initial reading, and it is computed using the following formula:

$$\text{GFI} = 0.4(\frac{\text{totalWords}}{\text{totalSentences}} + 100\frac{\text{longWords}}{\text{totalSentences}}),$$

where longWords are words longer than 7 characters. Higher values indicate lower readability.

The Automated Readability Index (ARI) (Smith and Senter, 1967) is an alternative readability formula that provides values correlating to the number of years of education needed to comprehend a given text:

$$\text{ARI} = 4.71(\frac{\text{totalCharacters}}{\text{totalWords}}) + 0.5(\frac{\text{totalWords}}{\text{totalSentences}}) - 21.43.$$

| Computing Infrastructure | 32GB NVIDIA V100 GPU |
|---|---|
| Optimizer | Adam |
| Optimizer Params | $\beta = (0.9, 0.999), \epsilon = 10^{-8}$ |
| learning rate | 1e-4 |
| Learning Rate Decay | Linear |
| Weight Decay | 0 |
| Warmup Steps | 0 |
| Maximum Gradient Norm | 1 |
| batch size | 4 |
| beam size | 3 |
| epoch | 10 |
| Lookahead $w$ | 25 |
| Lookahead $n$ | 20 |

Table 9: Hyperparameter settings for Flan-T5 methods.

Dale-Chall readability formula (Dale and Chall, 1948) (DCR) necessitates a compilation of 3000 words that are deemed understandable by fourth-grade students in the US. Words not found in this list are classified as difficult. The following expression is used in calculation:

$$\text{DCRF} = 0.1579(\frac{\text{difficultWords}}{\text{totalWords}} * 100) + 0.0496(\frac{\text{totalWords}}{\text{totalSentences}}).$$

Coleman-Liau index (CLI) (Coleman and Liau, 1975) relies on characters instead of syllables per word:

$$\text{CLI} = 0.0588L - 0.296S - 15.8,$$

where $L$ is the average number of letters and $S$ is the average number of sentences.

## B  Hyper-parameter Settings

The experiments were executed using the version 3.3.1 of the *transformers* library released by Hugging Face (Wolf et al., 2019). The fine-tuning process is halted once the model reaches convergence in terms of ROUGE score on the validation set. In Table 9, we report the hyperparameters used to train the models on CNN/DM. We use the Adam optimizer (Kingma and Ba, 2015) and employ a linearly decreasing learning rate schedule without warm-up.

## C  Experiments with GPT3.5

All of our experiments were conducted on the version of GPT3.5 (TEXT-DAVINCI-003) between 25 May 2023 and 13 Jun 2023 by using the OpenAI's API.[10] We set temperature = 1, top_p=1, frequency penalty = 0, and presence penalty = 0. For Best-GPT3.5, we select $k = 3$.

We select the the following prompts that gave the best results in terms of readability metrics:

---

[10]https://platform.openai.com/overview

| Readability level | DCR ↓ | ARI ↓ |
|---|---|---|
| GPT3.5 | | |
| 11-year-old | 9.9 | 12.5 |
| Middle school | 10.1 | 13.3 |
| High school | 10.4 | 14.1 |
| College student | 10.6 | 14.8 |
| Best-GPT3.5 | | |
| 11-year-old | 9.5 | 11.4 |
| Middle school | 9.8 | 12.3 |
| High school | 10.1 | 13.3 |
| College student | 10.7 | 15.5 |
| CATEGORYINSTRUCT | | |
| 11-year-old | 9.2 | 7.1 |
| Middle school | 9.7 | 8.8 |
| High school | 10.2 | 10.2 |
| College student | 10.8 | 11.4 |
| SCOREINSTRUCT | | |
| 90 (11-year-old) | 9.0 | 6.5 |
| 70 (Middle school) | 9.6 | 8.4 |
| 50 (High school) | 10.2 | 10.1 |
| 30 (College student) | 10.8 | 11.2 |
| SCOREINSTRUCT+RL | | |
| 90 (11-year-old) | 8.0 | 4.7 |
| 70 (Middle school) | 8.9 | 6.9 |
| 50 (High school) | 9.7 | 8.6 |
| 30 (College student) | 10.5 | 10.1 |
| SCOREINSTRUCT+LA | | |
| 90 (11-year-old) | 8.2 | 4.6 |
| 70 (Middle school) | 9.1 | 7.3 |
| 50 (High school) | 10.2 | 9.9 |
| 30 (College student) | 12.4 | 14.8 |

Table 10: Comparison on CNN/DM using instruction-based, RL and Lookahead (LA) methods using additional readability metrics. Lower DCR and ARI (↓) denote higher readability.

- {Document} \n Summarize the above article in 3 sentences for a sixth-grade student.

- {Document} \n Summarize the above article in 3 sentences for a middle-school student.

- {Document} \n Summarize the above article in 3 sentences for a high-school student.

- {Document} \n Summarize the above article in 3 sentences for a college student.

## D Results with Additional Readability Metrics

Table 10 shows results using additional metrics, Dale-Chall readability and Automated Readability Index for readability assessment. CATEGORY-INSTRUCT and SCOREINSTRUCT methods perform similarly with SCOREINSTRUCT's improvements in the *11-year-old* level. SCOREINSTRUCT+LA is able to better distinguish between readability levels compared to SCOREINSTRUCT+RL.

## E Mechanical Turk Setup

We provide additional details on our Amazon Mechanical Turk setup. AMT annotators are non-experts, so we use several mitigation strategies to obtain high-quality human judgements, including simplified task setups, clear annotation guidelines, task-specific qualification tests, and time checks to exclude potential spammers. We gave annotators fair compensation.

We give detailed instructions to the annotators, see Figures 6 and 7. We add a number of tasks with known answers (i.e., cases where the most/least readable/informative summaries should be clear), enabling us to estimate in real time the accuracy of workers who work on multiple of these. Workers who complete the tasks too quickly or have low accuracy on the tasks with known answers are automatically removed from our worker pool; their answers are replaced with new answers. We also use a bonus incentive structure. Every worker who passes the automatic quality checks receives a bonus at the end. In addition, we use custom qualification tests. For any worker to be accepted as an annotator for our readability and informativeness evaluations, there are three hurdles: (1) We only consider workers from a country whose main language is English, who has completed 100 or more HITs so far with an acceptance rate of 95% or higher. (2) In addition, workers must have passed an initial custom qualification test for a related text classification task we have conducted in the past. (3) The workers who have passed (1) and (2) qualify to take the custom qualification tests for our readability task and our informativeness task. Only the workers who passed these final tests were accepted to work on the human readability or informativeness evaluations in this paper.

On our batches of 1,000 HITs, we allowed any worker to complete a maximum of 333 HITs, so that no worker can dominate the results. We use two annotators per HIT.

Even though we only display three summaries at a time and only receive an annotated relative ranking of these three – as opposed to an absolute ordinal readability level – we wish to estimate a readability score for each of the four methods (Sec. 3) and their various target readability scores $\hat{r}$, so that these setups can be compared based on human judgements. We interpret each set of three summaries as a three-player game in which the method that generated the most readable summary

| | FRE↑ | GFI↓ | CLI↓ |
|---|---|---|---|
| **Document:** Team-mates Neymar and Dani Alves proved their dedication to Barcelona by supporting the club's basketball side. Neymar and Alves headed to watch El Clasico on Thursday night alongside the Brazilian's sister Rafaella. Barca prevailed with a narrow 85-80 victory in the Euro League contest. Brazil star Neymar (centre) takes a selfie with friends and Barcelona team-mate Dani Alves (right) However Real Madrid remain top of their Euro League division over their bitter rivals, just by points difference. Neymar helped Brazil beat Chile 1-0 at the Emirates Stadium on Sunday in a feisty contest and had to withstand several brutal challenges from the South American rivals. Before the international break Luis Enrique's Barcelona had won their El Clasico contest to move four points clear at the top of La Liga. Neymar and his Barcelona team-mates return to La Liga on Sunday, taking on Celta Vigo as they continue to compete for a treble of trophies. Neymar's sister Rafaella (left) headed to watch El Clasico of basketball with the Barcelona forward . Neymar's sister Rafaella (right) attends a Euro League basketball match between Barcelona and Real Madrid . Neymar is distracted at the basketball as his sister Rafaella (centre) watches on . Neymar, Brazil's captain, led his national side to a 1-0 win over Chile at the Emirates last weekend . Barcelona team-mate Dani Alves (front left) joined Neymar and friends to watch some live basketball . Gary Medel walks off having appeared to stamp on Neymar at the Emirates on Sunday . | 45.7 | 11.1 | 10.5 |
| **CATEGORYINSTRUCT** | | | |
| **Requested level: 11-year-old.** Neymar and Dani Alves went to watch basketball with their sister Rafaella. Barca beat Real Madrid 85-80 in the Euro League contest. Real remain top of the league by points difference. | 70.8 | 10.2 | 6.7 |
| **Requested level: middle school.** Neymar and Dani Alves watched basketball with Neymar's sister Rafaella. Barcelona beat Real Madrid 85-80 in the Euro League on Thursday. Real remain top of the league by points difference. Neymar helped Brazil beat Chile 1-0 at the Emirates on Sunday. | 58.1 | 10.6 | 6.0 |
| **Requested level: high school.** Neymar and Dani Alves watched basketball with Neymar's sister Rafaella. Barcelona beat Real Madrid 85-80 in the Euro League contest. Real remain top of the division over their bitter rivals by points difference. | 52.1 | 10.8 | 8.0 |
| **Requested level: college.** Neymar and Dani Alves attended El Clasico basketball match. Barcelona beat Real Madrid 85-80 in the Euro League contest. Neymar helped Brazil beat Chile 1-0 at the Emirates Stadium on Sunday. | 51.7 | 13.3 | 6.7 |
| **SCOREINSTRUCT** | | | |
| **Requested FRE 90.** Neymar and Dani Alves watched basketball with Neymar's sister Rafaella. Barca beat Real Madrid 85-80 in the Euro League on Thursday night. Real remain top of their division over their bitter rivals by just points difference. | 60.7 | 9.9 | 8.1 |
| **Requested FRE 70.** Neymar and Dani Alves watched basketball with Neymar's sister Rafaella. Barcelona beat Real Madrid 85-80 in the Euro League on Thursday night. Real remain top of their division over their bitter rivals by just points difference. | 56.0 | 9.9 | 8.1 |
| **Requested FRE 50.** Neymar and Dani Alves watched basketball with Neymar's sister Rafaella. Barcelona beat Real Madrid 85-80 in the Euro League on Thursday night. Real remain top of their division over Barcelona by just points difference. | 45.5 | 10.4 | 8.9 |
| **Requested FRE 30.** Neymar and Dani Alves watched basketball with Neymar's sister Rafaella. Barcelona beat Real Madrid 85-80 in the Euro League basketball contest. Real remain top of their division over Barcelona by points difference. | 42.6 | 11.5 | 9.2 |
| **SCOREINSTRUCT+RL** | | | |
| **Requested FRE 90.** Neymar and Dani Alves watched El Clasico of basketball with Neymar's sister Rafaella. Neymar led his Brazil side to a 1-0 win over Chile at the Emirates on Sunday. Barca beat Real Madrid 85-80 in the Euro League on Thursday night. Ney mar's side are four points clear at the top of La Liga. | 72.5 | 10.7 | 6.1 |
| **Requested FRE 70.** Neymar and Dani Alves watched basketball with Neymar's sister Rafaella at El Clasico on Thursday night. Neymar led Brazil to a 1-0 win over Chile at the Emirates Stadium on Sunday. Barcelona beat Real Madrid 85-80 in the Euro League basketball match. Ney mar's side are four points clear at the top of La Liga. | 65.2 | 11.2 | 6.9 |
| **Requested FRE 50.** Neymar and Dani Alves watched basketball with Neymar's sister Rafaella at El Clasico on Thursday night. Barcelona beat Real Madrid 85-80 in the Euro League basketball match at the Emirates Stadium. Neymar led Brazil to a 1-0 win over Chile at the weekend. | 54.5 | 12.0 | 7.5 |
| **Requested FRE 30.** Neymar and Dani Alves watched basketball with Neymar's sister Rafaella at El Clasico on Thursday night. Barcelona beat Real Madrid 85-80 in the Euro League basketball match at the Emirates Stadium. Neymar led Brazil to a 1-0 win over Chile at the weekend. | 54.5 | 12.0 | 7.5 |

Table 11: Examples of generated summaries for different readability levels.

wins and the method that generated the least readable summary loses. 22 workers worked on our readability evaluation, while 28 worked on our informativeness evaluation.

## F  Examples of Generated Summaries

Tables 11 and 12 present summaries of distinct levels generated by the different methods and their readability scores given by Flesch reading ease (FRE), Gunning fog index (GFI), and Coleman-Liau index (CLI) metrics.

| | FRE↑ | GFI↓ | CLI↓ |
|---|---|---|---|
| **Document:** A group of U.S. senators has written to football's world governing body FIFA, calling for Russia to be removed as host of the 2018 World Cup because of its role in the Ukraine crisis and occupation of Crimea. In a letter dated Tuesday and released on Wednesday, the 13 Democratic and Republican U.S. lawmakers said they 'strongly encourage' FIFA to move the global competition. 'Allowing Russia to host the World Cup inappropriately bolsters the prestige of the (Russian President Vladimir) Putin regime at a time when it should be condemned and provides economic relief at a time when much of the international community is imposing economic sanctions,' the senators wrote to FIFA President Sepp Blatter. FIFA president Sepp Blatter announces Russia as the host nation for the 2018 World Cup . Republican John McCain is one of 13 US senators urging FIFA to think again . The lead signers were Senator Robert Menendez, the top Democrat on the Senate Foreign Relations Committee, and Senator Ron Johnson, the Republican chairman of the foreign relations panel's Europe subcommittee. Republican Senator John McCain, chairman of the Senate Armed Services Committee, also signed the letter, as did Senator Richard Durbin, the number two Democrat in the Senate. England are planning a bid to host Eur 2028 after last holding the tournament at home in 1996 . Meanwhile, England are interested in bidding for Euro 2028 but are very unlikely to bid for the 2026 World Cup, FA chairman Greg Dyke has revealed. FIFA announced last week that the decision on the 2026 tournament will be made in 2017 - when Blatter is still likely to be FIFA president. The FA has decided not to bid for FIFA tournaments while Blatter is still in power following England's failed bid for 2018, but is eyeing a bid for the European Championship in 2028. | 40.5 | 11.6 | 15.3 |

**CATEGORYINSTRUCT**

| | FRE↑ | GFI↓ | CLI↓ |
|---|---|---|---|
| **Requested level: 11-years-old.** 13 US senators have written to FIFA asking for Russia to be removed as host of the 2018 World Cup. They say the decision on the 2026 World Cup will be made in 2017. England are planning a bid to host Euro 2028 but are unlikely to bid for the 2026 World Cup. | 86.7 | 8.3 | 10.0 |
| **Requested level: middle school.** 13 Democratic and Republican US senators have written to FIFA. They are calling for Russia to be removed as host of the 2018 World Cup. The letter was signed by Republican Senator John McCain. England are interested in bidding for Euro 2028 but are unlikely to bid for the 2026 World Cup. | 65.1 | 9.1 | 8.2 |
| **Requested level: high school.** 13 Democratic and Republican US senators have written to FIFA. They 'strongly encourage' FIFA to move the global competition. The senators say Russia's role in the Ukraine crisis and occupation of Crimea should be condemned. | 37.8 | 12.3 | 11.5 |
| **Requested level: college.** 13 Democratic and Republican U.S. lawmakers 'strongly encourage' FIFA to move the global competition. The senators said allowing Russia to host the World Cup 'inappropriately bolsters the prestige of the (Russian President Vladimir) Putin regime' | 21.7 | 13.7 | 13.8 |

**SCOREINSTRUCT**

| | FRE↑ | GFI↓ | CLI↓ |
|---|---|---|---|
| **Requested FRE 90.** 13 US senators call for Russia to be removed as host of the 2018 World Cup. They say the decision should be made in 2017 when FIFA president Sepp Blatter is still likely to be in power. England are planning a bid to host Euro 2028 but are unlikely to bid for the 2026 World Cup. | 79.1 | 8.8 | 11.7 |
| **Requested FRE 70.** 13 Democratic and Republican U.S. senators write to FIFA. They say Russia's hosting the World Cup 'bolsters the prestige of the Putin regime' England are planning a bid to host Euro 2028 but unlikely to bid for 2026. | 61.2 | 11.5 | 9.8 |
| **Requested FRE 50.** 13 Democratic and Republican U.S. senators write to FIFA. They 'strongly encourage' FIFA to move the global competition. England are planning a bid to host Euro 2028 after last holding the tournament at home in 1996. | 59.4 | 11.5 | 9.2 |
| **Requested FRE 30.** 13 Democratic and Republican U.S. lawmakers urge FIFA to move the global competition. 'Allowing Russia to host the World Cup inappropriately bolsters the prestige of the (Russian President Vladimir) Putin regime,' they wrote. | 30.3 | 13.3 | 10.3 |

**SCOREINSTRUCT+RL**

| | FRE↑ | GFI↓ | CLI↓ |
|---|---|---|---|
| **Requested FRE 90.** 13 senators have written to FIFA calling for Russia to be removed as host of the 2018 World Cup. The 13 senators say the tournament should be moved because of its role in the Ukraine crisis. England are planning a bid to host Euro 2028 but are unlikely to bid for the 2026 World Cup. | 72.8 | 9.1 | 10.9 |
| **Requested FRE 70.** 13 senators have written to FIFA calling for Russia to be removed as host of the 2018 World Cup. The 13 Democratic and Republican lawmakers say they 'strongly encourage' FIFA to move the tournament. Russia were announced as the host nation for the tournament in Russia. England are planning a bid to host Euro 2028 but are unlikely to bid for 2026. | 62.8 | 10.2 | 11.3 |
| **Requested FRE 50.** 13 Democratic and Republican senators have written to FIFA president Sepp Blatter calling for Russia to be removed as host of the 2018 World Cup. The senators say Russia's role in the Ukraine crisis and occupation of Crimea should be condemned. England are planning a bid to host Euro 2028 but are unlikely to bid for 2026 World Cup. | 56.4 | 10.7 | 12.6 |
| **Requested FRE 30.** 13 Democratic and Republican senators have written to FIFA president Sepp Blatter calling for Russia to be removed as host of the 2018 World Cup. The senators say Russia's role in the Ukraine crisis and occupation of Crimea should be condemned. England are unlikely to bid for the 2026 World Cup after failing to qualify for 2018 tournament. | 50.1 | 11.1 | 13.9 |

**SCOREINSTRUCT+LA**

| | FRE↑ | GFI↓ | CLI↓ |
|---|---|---|---|
| **Requested FRE 90.** 13 U.S. senators call for Russia to be removed as hosts of the 2018 World Cup. They say the decision should be made in 2017. England are planning a bid to host Euro 2028 but are unlikely to bid for the 2026 World Cup. | 90.0 | 8.9 | 9.5 |
| **Requested FRE 70.** US senators call on FIFA to move the 2018 World Cup from Russia. Russia was announced as the host nation for the 2018 tournament. 13 Democratic and Republican U.S. senators signed the letter. | 65.9 | 11.1 | 9.2 |
| **Requested FRE 50.** FIFA president Sepp Blatter announced Russia as host nation for 2018 World Cup. 13 Democratic and Republican U.S. senators have written to FIFA president urging him to move the event to 2018. The lawmakers said Russia's role in the Ukraine crisis and occupation of Crimea should be condemned. | 50.4 | 13.1 | 12.2 |
| **Requested FRE 30.** 13 Democratic and Republican U.S. lawmakers urge football's world governing body FIFA to move the global competition. The senators said Russia hosting the 2018 World Cup 'inappropriately bolsters the prestige of the (Russian President Vladimir) Putin regime' England are interested in Euro 2028 but are very unlikely to bid for the 2026 World Cup, FA chairman Greg Dyke has revealed. | 30.2 | 13.4 | 17.4 |

Table 12: Examples of generated summaries for different readability levels measured using FRE, GFI and CLI metrics.

**Instructions** (Click to collapse)

**Welcome!**

We need your help on evaluating three **automatically generated** summaries.

For these three summaries, select the **most readable** and the **least readable**.

The most readable summary uses shorter, easier or more common words, less technical jargon, shorter sentences, less complex grammar. It is easier to read for younger readers.

The least readable summary uses some longer or uncommon words, more technical jargon, longer sentences, more complex grammar. It is harder to read for younger readers.

When you select one as most readable, it will turn **green**.

When you select one as least readable, it will turn **red**.

**Ties:** If summaries are tied you may select multiple summaries. You can select multiple *most readable* summaries or multiple *least readable* summaries. This makes sense when the readability of different summaries are very similar. You do NOT need to select multiple most readability (or multiple least readable) summaries every time. It should happen only in the case of ties. It is acceptable (and encouraged) to select *one* as most readable, *one* as least readable and leave one unchecked.

The three summaries on the page all summarize the same text. We display the summaries only, not the text that they are summarizing, to save you time and effort.

Note that the summaries occasionally describe distressing events, such as violent crimes.

**Please evaluate the three summaries shown below.**

Select the **most readable** summary (i.e., uses shorter, easier or more common words; less technical jargon, shorter sentences, less complex grammar. It is easier to read for younger readers.

Select the **least readable** summary (i.e., uses some longer or uncommon words; more technical jargon, longer sentences, or more complex grammar. It is harder to read for younger readers.

*Ties: You may select multiple if there are multiple very similar most readable or least readable summaries.*

| Most readable ↓ | Least readable ↓ | Summary |
|:---:|:---:|:---|
| ☐ | ☐ | ${summary1} |
| ☐ | ☐ | ${summary2} |
| ☐ | ☐ | ${summary3} |

Submit

Figure 6: Screenshot of the human readability annotation.

**Instructions** (Click to collapse)

**Welcome!**

We need your help on evaluating three **automatically generated** summaries by comparing them to the original article that the summaries try to summarize.

For these three summaries, you only need to select one as **most informative** and one as **least informative**.

The most informative summary is best at expressing the main points of the article, its content is the most important and relevant. It's not necessarily the longest summary -- just the one that contains the most important information.

The least informative summary is worst at expressing the main points of the article, its content is the least important and relevant. It's not necessarily the shortest summary -- just the one with the least important information.

When you select one as most informative, it will turn **green**.

When you select one as least informative, it will turn **red**.

**Ties:** If summaries are tied you may select multiple summaries. You can select multiple *most informative* summaries or multiple *least informative* summaries. This makes sense when different summaries have very similar informativeness. You do NOT need to select multiple most informativeness (or multiple least informative) summaries every time. It should happen only in the case of ties. It is acceptable (and encouraged) to select *one* as most informative, *one* as least informative and leave one unchecked.

Note that the texts occasionally describe distressing events, such as violent crimes.

**Please evaluate the three summaries shown below the article.**

First, here is the article:

${input}

Select one summary as the best, i.e., the *most* informative summary! That summary is best at expressing the main points of the article, its content is the most important and relevant.

Select one summary as the worst, i.e., the *least* informative summary! That summary is worst at expressing the main points of the article, its content is the least important and relevant.

*Ties: You may select multiple if there are multiple very similar most informative or least informative summaries.*

| Most informative ↓ | Least informative ↓ | Summary |
|---|---|---|
| ☐ | ☐ | ${summary1} |
| ☐ | ☐ | ${summary2} |
| ☐ | ☐ | ${summary3} |

Submit

Figure 7: Screenshot of the human informativeness annotation.