# OpenReview forum: "Generating Summaries with Controllable Readability Levels"
_EMNLP/2023/Conference — EMNLP 2023 Main_

### Official Review · Reviewer_NGJK · 2023-08-01

**Typos Grammar Style And Presentation Improvements:** 1. It is unclear why there would be a…
**Soundness:** 3

**Excitement:**

4: Strong: This paper deepens the understanding of some phenomenon or lowers the barriers to an existing research direction.

**Paper Topic And Main Contributions:**

This paper introduces a variant on the summarization task which looks at generating summaries at a given readability level

The authors propose 3 baseline methods that provide control over the readability of summaries:

- Instruction-prompting

- Reinforcing learning

- Lookahead decoding

The authors evaluate these methods including some combination of them and show that these methods can control the readability of the generated summaries

Finally, the authors analyze the tradeoff between readability and other aspects of summaries like specificity, abstractions, factuality and informativeness

**Questions For The Authors:**

1. What is the motivation for the GPT-3.5 baseline?

2. Language models often struggle with math and numeric heavy tasks and it seems like you are just encoding the requested score as any other text for “Score-based Instructions”. Have you done any validation to see if the model understands the numbers provided?

**Reasons To Accept:**

1. I believe that the tradeoff between readability and specificity offered by this variant of the summarization task would be useful
2. The proposed methods seem like they would be useful a foundation for future work on this task

**Reasons To Reject:**

1. I don’t think the comparison to GPT-3 is fair
It is evaluated very differently from other other models, so it is not possible to tell if the difference in performance is due to a difference in evaluation:

a. Other models are decoding using some sort of beam decoding (beam search?)
GPT-3 uses sampling with temperature=0, which should be equivalent to greedy search (greedy search is know to perform worse than beam search)

b. A different prompt is used for GPT-3 than for the Category-based Instructions method, were both prompts tried on both methods and the best performing prompt selected? (the difference would be sense in this case, but this is not clearly indicated in the paper)

I would consider trying to make this baseline more comparable to the proposed methods or excluding it

2. Motivation of work is weak
You motivate higher readability (“Higher readability lowers reading efforts and increases the speed for any reader”), but you do not motivate why lower readability would be desired (You do show that higher readability results in lower specificity, I would mention something about this in your motivation)

Edit: In the authors rebuttal, they include a results for GPT-3.5 with sampling which shows an improvement in performance, but still less than the proposed method.  The authors will also include the methods used to create prompts for this summarization task and include these in the paper's appendix. The authors said that they will make the motivation for the control of readability levels (higher readability levels leading to lower specificity and interestingness) more clear in the final version of the paper.

**Reproducibility:**

3: Could reproduce the results with some difficulty. The settings of parameters are underspecified or subjectively determined; the training/evaluation data are not widely available.

**Reviewer Confidence:**

4: Quite sure. I tried to check the important points carefully. It's unlikely, though conceivable, that I missed something that should affect my ratings.

---

> ### Author Rebuttal · Authors · 2023-08-28
>
> Thank you for your valuable feedback and for appreciating that our investigation of the tradeoff between readability and specificity is useful and for seeing our work as a foundation for future research in this direction.
>
> **- About GPT3.5 as baseline**
>
> Our motivation for adding the GPT3.5 baseline is that it is widely considered to be a strong text generation model that performs well on a multitude of NLP tasks based on instructions and achieves state-of-the-art results on summarization [1, (Line 044 - Line 047)] and controllable summarization [2, (Line 515 - Line 518)]. We believe that it is considered a strong model despite not using beam search. We addressed your comment and in order to mitigate this issue (GPT3.5 not using beam search), we generate multiple (3) summaries from GPT3.5 for each example and instruction and select the summary whose readability level is closest to the requested readability level (see BestGPT3.5 below; we experiment with 50 examples given the costs to run GPT3.5 on 3 * 4 readability levels * # examples):
>
> | | ΔFRE(↓) | FREρ(↑) | GFIρ(↓) | CLIρ(↓)
> | --- | ----------- |----------- | ----------- | ----------- |
> | GPT3.5 | 24.6|  0.23 |  -0.24|  -0.16
> | BestGPT3.5 | 21.1|  0.28 | -0.30 | -0.15
> | CategoryInstruct |  16.9 | 0.61 |  -0.40 |  -0.55
> | ScoreInstruct | 15.7 |  0.61 |  -0.41 |  -0.56
> | ScoreInstruct+RL | 12.2 | 0.74 | -0.42 |  -0.73
> | ScoreInstruct+LA | 5.4|  0.95 |  -0.65 |  -0.86
>
> As shown above, BestGPT3.5 improves over GPT3.5 but is still behind our methods in the readability control. Upon acceptance, we will include BestGPT3.5 as an additional baseline.
>
> **- About the differences in instructions between GPT3.5 and our methods**
>
> Following previous work [1], we used sentence-count length GPT3.5 prompts to adapt to CNN/DM dataset. We have experimented with many different instructions for readability control in preliminary experiments inspired by examples provided in the GPT models documentation and have selected the best-performing ones for each category. For example, for the category with highest readability (Flesch Reading Ease ≥80), we experimented with following instructions on CNN/DM 50 examples:
>
> **Instruction:** Summarize the above article in 3 sentences for a sixth-grade student.
> **Readability metrics:** Flesch Reading Ease: 43.9, Gunning Fog Index: 13.4, Coleman-Liau Index: 12.4
>
> **Instruction:** Summarize the above article in 3 sentences for a five-grade student.
> **Readability metrics:** Flesch Reading Ease: 39.3, Gunning Fog Index: 14.4, Coleman-Liau Index: 13.0
>
> **Instruction:** Summarize the above article in 3 sentences for an average 11-years-old student.
> **Readability metrics:** Flesch Reading Ease: 40.1, Gunning Fog Index: 13.9, Coleman-Liau Index: 13.1
>
> We selected the first instruction which gave the higher readability average scores for all metrics. We will describe all preliminary prompt variants in Appendix. Finally, our findings (Line 392 - Line 399) that GPT3.5 is not strong in fine-grained readability control are in line with recent findings by Pu and Demberg (2023).
>
> **- About the motivation**
>
> You suggest that we add better motivation for requesting lower readability levels (e.g., college level), e.g., by stressing earlier in the text that high readability levels correlate with lower specificity and interestingness; we will describe this point further and add it to the Motivation section. Additionally, our experiments are on a subset of CNN/DM articles that have lower readability to begin with (high school and college levels), in order to minimize cases where we request a summary with a lower readability level than the input article (Line 328 - Line 333).
>
>
> **- About the numerical scores in the instructions**
>
> It is true that LLMs tend to struggle with numerical tasks. However, we expect that the Flan-T5-Large model, which has close to 1B parameters and has been extensively pre-trained and fine-tuned (including arithmetic reasoning tasks [3]) and can learn meaningful representations of simple integer numbers under 100 [4] which is not a heavy mathematical task. Figure 4a presents evidence that our score-based methods are capable of generating summaries with different readability levels based on the distinct numerical scores presented in the instructions in a monotonic way. As shown in Table 2, we find strong Pearson correlations between the requested score in the instructions and the observed readability; for example, 0.62, 0.74 for ScoreInstruct and ScoreInstruct+RL, demonstrating that our ScoreInstruct methods are able to follow instructions based on numbers.
>
>
> **- About the presentation improvements**
>
> Thanks for the helpful feedback on the text and image improvements, which we will incorporate into the camera-ready version. We will move fair payment and other details about our Mechanical Turk setup to the Ethics statement.
>
> [1] News Summarization and Evaluation in the Era of GPT-3 (Goyal et at. 2023)
>
> [2] Exploring the Limits of ChatGPT for Query or Aspect-based Text Summarization (Yang et al. 2023)
>
> [3] Scaling Instruction-Fine Tuned Language Models (Chung et al. 2022)
>
> [4] Investigating Numeracy Learning Ability of a Text-to-Text Transfer Model (Pal & Baral, EMNLP Findings 2021)

---

### Official Review · Reviewer_MNBZ · 2023-08-02

**Typos Grammar Style And Presentation Improvements:** Figure 1
**Soundness:** 4

**Excitement:**

4: Strong: This paper deepens the understanding of some phenomenon or lowers the barriers to an existing research direction.

**Missing References:**

1. Flesch-Kincaid is Not a Text Simplification Evaluation Metric (Tanprasert & Kauchak, GEM 2021)

**Paper Topic And Main Contributions:**

This paper presents three approaches for generating summaries with readability control. This is done via using language models with instruction finetuning where the target readability levels are either presented in a categorical or in a score-based fashion. The score-based instruction models are further finetuned with reinforcement learning (RL) or are decoded using a look-ahead (LA) mechanism to achieve further control over the target readability. Results demonstrate that both RL and LA improve control over target readability compared to the base ScoreInstruct model.

**Questions For The Authors:**

See Reasons to Reject

**Reasons To Accept:**

1. Thorough automatic and human evaluation demonstrates the effectiveness of the proposed RL and LA approach.
2. Supporting ablation analysis shows how readability impacts the specificity and informativeness of the target summary.
3. Controlling aspects of summary beyond abstractiveness and factuality is relatively less explored and this paper makes an interesting contribution in that direction.

**Reasons To Reject:**

1. Readability control using FRE and other readability metrics while a reasonable approach is also easily tunable (see [1]). No qualitative analysis is provided to support whether the summaries generated for the 4 levels differ in an expected fashion (similar to what is shown in Figure 1). FRE scores are biased towards the shorter text and hence it's unclear whether the summaries are simpler or just shorter.
2. While readability control is less explored in summaries, controllable text simplification has received a lot of attention from the NLP community where the goal is to modify an input text with a desired U.S. reading grade level or control other aspects of readability/simplicity. No discussion of existing literature on controllable simplification is presented in prior work.

In the Rebuttal period, the author provided appropriate responses for points 1 and 2. I would encourage the authors to include the example and discussion of Point 1 in the camera ready version.

**Reproducibility:**

4: Could mostly reproduce the results, but there may be some variation because of sample variance or minor variations in their interpretation of the protocol or method.

**Reviewer Confidence:**

5: Positive that my evaluation is correct. I read the paper very carefully and I am very familiar with related work.

---

> ### Author Rebuttal · Authors · 2023-08-28
>
> Thank you for your valuable feedback and positive comments, and for appreciating our automatic and human evaluations and ablation studies, and that controlling aspects beyond abstractiveness and factuality has been an understudied area and interesting contribution.
>
> **- About readability control using FRE**
>
> In our experiments, we find that optimizing toward higher FRE readability scores leads to summaries that contain words that are shorter and easier to read and sentences that are shorter. We do not observe that it leads to very short summaries overall. Importantly, note that (i) instruction-based methods are trained on reference summaries; (ii) the PPO method incorporates a KL penalty between the initial policy and the updated policy making sure the outputs look like CNN/DM summaries; and (iii) the Lookahead method is controlled by a weight in the score function (Equation 4, Line 311 - Line 313), which mitigate such issues.  In what follows, we present the average length of the generated summaries for different levels:
>
> |      | 11-year-old | middle school | high school| college|
> | ----------- | ----------- |----------- |----------- |----------- |
> | CategoryInstruct | 43.6 | 49.4 |   51.5 |   46.4
> | ScoreInstruct | 35.2 | 46.9 |  47.5 |  42.6
> | ScoreInstruct+RL | 55.9|  56.3 | 54.8 | 52.5
> | ScoreInstruct+LA | 34.5 |  42.5 | 46.7 | 40.0
>
> There is a slight difference in terms of number of words between the more readable and less readable summaries, although the higher readable summaries are not very short. Below we give an example of summaries with the lengths for different readability levels:
>
> **Document:**
> …which Premier League club is the most popular according to Twitter followers. For the first time ever, fans can see a detailed breakdown of how support for every club varies around the world and in the UK supporters can go as in-depth as seeing the three most popular teams in their local constituency…. Liverpool may be struggling … but in terms of Twitter followers they dominate the UK. Arsenal have strong support … Chelsea and Tottenham. In Manchester, it is a win for Louis van Gaal's …
>
> **Requested Flesch Reading Ease score: 90 (easily understood by an 11-year-old student).**
> Summary: Premier League Twitter followers are the new tool to help fans make up their mind. The map shows which club is most popular in each country in the world. Liverpool has the most followers in the UK, but is less popular in other areas. Arsenal are the most popular team in the world with 5.6m followers.
> **Length:** 56, **Observed readability scores:** Flesch Reading Ease (higher more readable): 76.5, Gunning Fog Index (lower more readable): 10.2, Coleman-Liau Index (lower more readable): 7.0.
>
> **Requested Flesch Reading Ease score: 70 (Plain english, fairly easy to read).**
> **Summary:**  Supporters can view a breakdown of how each club is supported around the world. Liverpool dominates the UK market but has less support in other areas. Chelsea and Manchester United are the most popular in Asia apart from Thailand. Arsenal are the most popular team in the world with 5.6million followers.
> **Length:** 51, **Observed readability scores:** Flesch Reading Ease: 61.2, Gunning Fog Index: 9.5, Coleman-Liau Index: 10.0.
>
> **Requested Flesch Reading Ease score: 50 (fairly difficult to read).**
> **Summary:**  The new Twitter map was created by looking at the official Twitter accounts for every team, using their followers as an indicator of allegiance. Liverpool dominates the UK market but is less popular in other parts of the world. Arsenal is the most popular team with 5.6million followers, followed by Chelsea (5.4m) and Manchester United (4.8m) across the globe, Arsenal is the most popular team with 5.6million followers, followed by Chelsea (5.4m) and Manchester United (4.8m)
> **Length:** 76, **Observed readability scores:** Flesch Reading Ease: 53.6, Gunning Fog Index: 12.6, Coleman-Liau Index: 8.5.
>
> **Requested Flesch Reading Ease score: 30 (difficult to read).**
> **Summary:**  Supporters of the Premier League can view how their club's global popularity differs. Liverpool dominate UK Twitter following, followed by Arsenal, Chelsea and Manchester United. Supporters in the UK can also see their favourite team in their local constituency.
> **Length:** 39, **Observed readability scores:** Flesch Reading Ease: 33.1, Gunning Fog Index: 11.3, Coleman-Liau Index: 14.9.
>
> Finally, we will cite and discuss the missing reference and add an analysis of the summary length size along with example summaries in the final version given the extra page.
>
>
> **- About controllable text simplification**
>
> Thank you for your insightful suggestion. We will include and discuss related work on text simplification and point out the differences to our work in the final version. We note that in contrast to controllable simplification approaches (which aim to control the degree to which a sentence is paraphrased), controllable summarization approaches must provide succinct and informative summaries while maintaining the desired reading level (Line 135 - Line 139).

---

### Official Review · Reviewer_RnEB · 2023-08-03

**Soundness:** 4

**Excitement:**

3: Ambivalent: It has merits (e.g., it reports state-of-the-art results, the idea is nice), but there are key weaknesses (e.g., it describes incremental work), and it can significantly benefit from another round of revision. However, I won't object to accepting it if my co-reviewers champion it.

**Paper Topic And Main Contributions:**

This paper proposes to generate text summarization with controlled readability. First the proposed method prompts summaries with different readability from LLM. Then it tunes the policy model with PPO to achieve controllable ability. The method is evaluated on CNN Dailymail, and shows better results than GPT 3.5.

**Questions For The Authors:**

1. For score based instruction, what does the score here mean? When you want to prompt a summary, what scores would you use? Would the space be too sparse as good summarizations are limited for each document, but the score is a real-value ranging from 0 to 100, say a large space.  I guess the <summary, score> pair space would be very sparse.
2. Can author show more cases about the generated summary in different readability level?

**Reasons To Accept:**

The paper is well-written and well-organized, making it easy to follow.

**Reasons To Reject:**

1. The proposed method seems not very new. Prompting from LLM to do data augmentation, and applying PPO both are very common techniques.
2. The experimental settings might have some issues. For readability, a older person should understand all the contents within or lower his/her age range, which seems more reasonable. However, here the authors limited the people of given age can only understand the corresponding summaries. It seems unreasonable for me,

**Reproducibility:**

4: Could mostly reproduce the results, but there may be some variation because of sample variance or minor variations in their interpretation of the protocol or method.

**Reviewer Confidence:**

4: Quite sure. I tried to check the important points carefully. It's unlikely, though conceivable, that I missed something that should affect my ratings.

---

> ### Author Rebuttal · Authors · 2023-08-28
>
> Thank you for your valuable feedback and for noting that our paper is well-written and well-organized.
>
> **- About PPO being a common technique**
>
> We would like to address your point about the novelty of our paper. While using PPO to optimize toward a reward function is common, our specific reward function (Line 256 - Line 272) is an innovation over the standard PPO, and we designed it for our task of fine-grained control of reading levels requested at inference time. Generally, the reward is given directly by a reward model [1, 2], but in our paper the reward is computed as the (non-linear Gaussian) distance of the observed readability to the requested readability (Line 263 - Line 264). More details about this setting are described in Section 4 (Line 343 - Line 251). This setup introduces flexibility and controllability at inference time that is absent in typical PPO-RLHF papers [3]. PPO is usually used in the context of human feedback (i.e., RLHF), but our approach does not require learning a reward model from human feedback.
>
> **- About the experimental settings**
>
> Readability levels are not determined solely by age, but by educational level. We followed established guidelines for text complexity levels from previous work (Line 153 - Line 159 and Line 194 - Line 199) using (US) grade levels of education. The summary can be understood by someone at the complexity level and below. Moreover, while our proposed methods (CategoryInstruct, ScoreInstruct) are flexible and can encode different readability definitions via instructions – for example the ones based on Common European Framework of Reference for Languages (CEFR) levels [4], we focus on (US) grade levels which are used to estimate the years of formal education using the different readability metrics described in our submission (Appendix A - Line 964 - Line 995). We will provide additional clarification on the levels of education in the final version.
>
> **- About the score-based instructions**
>
> For our method with score-based instructions (ScoreInstruct), we employ the Flesch Reading Ease (FRE) metric as the score in the experiments (Line 334 - Line 338). Table 1 shows the mapping between the FRE scores and the intended readability level described in the prompts, based on previous work [5]. For example, a summary with a FRE score of 80 or higher is easy and quick to read. In contrast, a text with a FRE score smaller than 40 is difficult to read and best understood by college graduates. For a given readability level, a prompt instruction with a score comprehending such level (Table 1) can be used.
> You ask if the score-based instruction is sparse. The specific scores in the prompt are indeed sparse. However, we expect that the Flan-T5-Large model, which has close to 1B parameters and has been extensively pre-trained and fine-tuned on different tasks (including arithmetic reasoning [6]). Previous work [7] found that indeed T5 models can learn meaningful representation for integer numbers. Furthermore, Figure 4a shows that our score-based methods learn to generate summaries with different readability levels based on the distinct numerical scores presented in the instructions. Finally, Table 2 shows that the Pearson correlations between the requested score in the instructions and the observed readability are strong: 0.62, 0.74 for ScoreInstruct and ScoreInstruct+RL;, demonstrating that our approaches are able to follow the instructions with numerical values. We will include further details about ScoreInstruct in the final version.
>
> **- About summary examples with different readability levels**
>
> Below we present examples of summaries with distinct readability levels generated by the lookahead method:
>
> **Document:**
> An 18-year-old man has been charged with conspiring to commit terrorist acts following Saturday morning's counter-terror police raids, in which police seized a number of knives and swords. Sevdet Besim … was charged with 'conspiracy to commit acts done in preparation for, or planning, terrorist acts' at the Melbourne Magistrates Court….
>
> **Requested Flesch Reading Ease score: 90 (easily understood by an 11-year-old student).**
> **Summary:** A total of five teens are being held by police for 'simply evil' plans. One man was charged with terrorism and will be remanded in custody. He is one of two men, both 18, who were arrested in the raids. Police seized a number of knives and swords at some of the homes.
> **Observed readability scores:** Flesch Reading Ease (higher more readable): 81.4, Gunning Fog Index (lower more readable): 9.0, Coleman-Liau Index (lower more readable):  5.5.
>
> **Requested Flesch Reading Ease score: 70 (Plain english, fairly easy to read).**
> **Summary:** A total of five men were arrested by police as part of Operation Rising on Saturday morning. One man has been charged with conspiring to commit terrorist acts and is due to appear in court on April 24. Sevdet Besim, 18, did not apply for bail and was remanded in custody for a filing hearing on April 24. Two of the men were 'associates' of Abdul Numan Haider, the 18-year-old who was shot dead on September 23. Police seized a number of 'edged weapons' including knives and a sword.
> **Observed readability scores:** Flesch Reading Ease: 62.3, Gunning Fog Index: 10.2, Coleman-Liau Index: 7.7.
>
> **Requested Flesch Reading Ease score: 50 (fairly difficult to read).**
> **Summary:** Sevdet Besim, 18, was charged with 'conspiracy to commit acts done in preparation for, or planning, terrorist acts' on Sunday afternoon. He is one of two 18-year-old men arrested following counter-terror police raids in Melbourne. The men were 'associates' of Abdul Numan Haider, who was shot dead after stabbing two officers outside a Melbourne police station on September 23. The second teenager is still being interviewed by police who plan to charge him with a number of offences related to preparing for and planning terrorist acts.
> **Observed readability scores:** Flesch Reading Ease: 44.7, Gunning Fog Index: 14.6, Coleman-Liau Index: 12.4.
>
> **Requested Flesch Reading Ease score: 30 (difficult to read).**
> **Summary:**  Sevdet Besim was charged with 'conspiracy to commit acts done in preparation for, or planning, terrorist acts'. He's one of two 18-year-old men arrested for allegedly planning an 'ISIS inspired' attack on an Anzac Day ceremony. The men were 'associates' of Abdul Numan Haider, the 18-year-old gunman who was shot dead after stabbing two officers outside a Melbourne police station on September 23. Victorian Premier Daniel Andrews dubbed the plans 'simply evil'.
> **Observed readability scores:** Flesch Reading Ease: 34.9, Gunning Fog Index: 15.2, Coleman-Liau Index: 13.1.
>
> We will include examples of generated summaries of various reading levels for different methods to the Appendix in the paper.
>
>
> [1] Better Rewards Yield Better Summaries: Learning to Summarise Without References (Böhm et al., EMNLP-IJCNLP 2019)
>
> [2] Recursively Summarizing Books with Human Feedback (Stiennon et al., NeurIPS'20)
>
> [3] Learning to summarize from human feedback (Wu et al., NeurIPS'20)
>
> [4] Learning to Paraphrase Sentences to Different Complexity Levels (Chi et al., 2023)
>
> [5] How to Write Plain English (Flesch, Rudolf. University of Canterbury. 1979)
>
> [6] Scaling Instruction-Fine Tuned Language Models (Chung et al. 2022)
>
> [7] Investigating Numeracy Learning Ability of a Text-to-Text Transfer Model (Pal & Baral, EMNLP Findings 2021)

---

### Meta-Review · Area_Chair_JLnX · 2023-09-19

**Recommendation:** 5

**Metareview:**

The paper explores methods for generating text summaries with controlled readability. It is well-written and well-organized. The effectiveness of the proposed RL and LA approach is demonstrated through comprehensive automatic and human evaluations. The tradeoff between readability and specificity in this summarization task also holds promise. While promising, reviewers have revised some moderate concerns about the motivation for controlling readability levels, the novelty of the proposed method, and issues with the experimental setup.

---

### Decision · Program_Chairs · 2023-10-07

**Decision:**

Accept-Main

**Comment:**

The paper explores methods for generating text summaries with controlled readability. It is well-written and well-organized. The effectiveness of the proposed RL and LA approach is demonstrated through comprehensive automatic and human evaluations. The tradeoff between readability and specificity in this summarization task also holds promise. While promising, reviewers have revised some moderate concerns about the motivation for controlling readability levels, the novelty of the proposed method, and issues with the experimental setup.